# Perceived Satisfaction with Online Study during COVID-19 Lockdown Correlates Positively with Resilience and Negatively with Anxiety, Depression, and Stress among Slovenian Postsecondary Students

**DOI:** 10.3390/ijerph19127024

**Published:** 2022-06-08

**Authors:** Branko Gabrovec, Špela Selak, Nuša Crnkovič, Katarina Cesar, Andrej Šorgo

**Affiliations:** 1National Institute of Public Health, Trubarjeva Cesta 2, 1000 Ljubljana, Slovenia; spela.selak@nijz.si (Š.S.); nusa.crnkovic@nijz.si (N.C.); katarina.cesar@nijz.si (K.C.); 2Faculty of Natural Science and Mathematics, University of Maribor, Koroška Cesta 160, 2000 Maribor, Slovenia; andrej.sorgo@um.si

**Keywords:** online study, COVID-19, resilience, anxiety, depression

## Abstract

Background: The purpose of this study is to fill the research gap regarding the influence of satisfaction with distance learning on the correlates of mental health during the COVID-19 pandemic. Methods: An online cross-sectional study was conducted in February and March 2021, involving 4661 postsecondary students. Five validated instruments—PHQ-9 (depression), GAD-7 (anxiety), PSS-4 (stress), CD-RISC-10 (resilience) and SAT-5 (satisfaction with online study)—were used in the present study. Findings: The correlations between anxiety, depression, and stress were so high that they were almost inextricably linked. Both satisfaction with online learning and psychological resilience were negatively correlated with anxiety, depression, and stress. Satisfaction with online learning was also negatively correlated with psychological resilience. Females showed higher levels of vulnerability to anxiety, depression, and stress, and exhibited lower levels of psychological resilience than males. Conclusion: Home-based distance-learning during the COVID-19-induced lockdown had a significant impact on students’ mental health. Low satisfaction with distance learning can lead to the development of anxiety and depression symptoms, increase stress, and decrease the psychological resilience of postsecondary students; therefore, it is critical that educational institutions focus on implementing interventions that promote students’ satisfaction with distance learning, and their psychological resilience, to protect their mental health.

## 1. Introduction

The COVID-19 pandemic caused unprecedented disruption, and the response of governments and local authorities was far from coordinated in following the best practices based on current evidence [1]. The pandemic had an impact on various aspects of life, including the mental health of postsecondary students [2]. Higher education institutions were suspended in 188 countries worldwide, and teaching has been shifted to home-based distance learning models to control the spread of the COVID-19 pandemic [3]. Although several studies have attempted to assess the actual effectiveness of school closures for pandemic control over time, no definitive answer to this question has been provided [4]. While much has been discussed about the challenges faced by faculties and their institutions during the COVID-19 pandemic [5], less has been reported about how students coped with the challenges [6], and even less has been reported about the relationship between satisfaction with forced online education, psychological resilience, and students’ mental health.

During a pandemic, many stressful events occur that can affect mental health [7,8]. Although stress is a normal response to stressful circumstances, the effects on mental health occur when the stress is present for too long and is too intense to be successfully managed [9]. Previous research reported strong associations between psychological stress, poor academic performance, and career outcomes [10]. Moreover, female students appear to be less successful in coping with stress compared to male students [11]. As both the education process and the pandemic are recognized as a high-stress periods, post-secondary students have been at even greater risk of developing mental health difficulties during the COVID-19 pandemic, especially female students [12,13]. This increased risk was reflected in the higher prevalence of mental health difficulties among postsecondary students. For example, researchers from European countries reported a high prevalence of depressive symptoms within the student population, ranging from 27.2% to 43.0% [14], [15], with 2.2% to 7.0% experiencing severe depressive symptomatology. Moreover, in Italy, more than 50% of participants reported symptoms of anxiety during the COVID-19 lockdown [16]; French university students reported moderate (51.7%) to severe (22.0%) stress [17]; and in Poland, 56% of university students reported high or extremely high levels of stress [15]. Both of the latter studies reported that female students exhibited higher levels of mental health difficulties in comparison to their male peers. Data from a Slovenian study showed that the symptomatology of depression (assessed using the Patient Health Questionnaire (PHQ-8)) was quite high among students participating in the survey (M = 11.36; SD = 6.27). Moreover, almost all students participating in the study noticed, at least sometimes, certain signs of depression and anxiety, felt a negative attitude towards themselves, and struggled with chronic fatigue/exhaustion [18].

Because stress, anxiety, and depression can occur in response to adverse events such as a pandemic and have negative effects on everyday functioning (e.g., academic performance, intimate relationships, athletic performance) [19,20], it is critical to identify factors that could protect individuals from developing mental health problems when faced with the aforementioned events. One of the protective psychological constructs that is highly relevant to the COVID-19 pandemic is psychological resilience; it has been associated with positive outcomes in a number of domains (e.g., education, occupation, sport) and provides a conceptual framework for understanding individual differences in resilience levels, which appears to be a critical factor in promoting good psychological functioning during the COVID-19 pandemic [21]. Moreover, psychological resilience has been shown to be negatively associated with mental health problems such as higher levels of stress, anxiety, and depression [19,22].

To the best of our knowledge, no research has examined how different mental health variables correlate with satisfaction with distance learning among post-secondary students during the COVID-19 pandemic, while also focusing on gender differences; therefore, this was explored in the present study. This was carried out in three steps, starting with the examination of the correlations among anxiety, depression, stress, psychological resilience, and satisfaction with distance learning. This was followed by examining gender differences in relation to the aforementioned constructs. Lastly, a comparison between different groups of students based on their level of reported satisfaction with distance learning in relation to mental health constructs was examined. All hypotheses can be summarized in the assertion that all of the above constructs are interrelated, but no presumptions as to whether the associations are negative or positive were made in advance.

## 2. Materials and Methods

### 2.1. Participants

Because substantial efforts were made to reach the entire tertiary student population, a call to participate was announced on multiple institutional webpages so that most students had an opportunity to respond. The authors of the study hoped that the guaranteed anonymity would give students freedom to provide candid responses. A possible source of bias was self-election, and the difference between the population of students who responded and those who did not is unknown.

The questionnaire was completed by 7154 student participants, of which 83.86% (*n* = 5999) were full-time postsecondary students. To perform comparative analysis based on the sum scores of PHQ-9, GAD-7, PSS-4, CD-RISC-10, and SAT, participants for whom data or any of the items of the scales were missing were removed. Thus, we were left with 4661 participants. Most participants were female (72.5%), with 26.7% being male and 0.8% identifying as another gender. The average age of the participants was 22.85 years, with the youngest being 17 years old and the oldest being 62 years old. Most of the participants were enrolled in the Bachelor’s and Single Cycle Master’s degree programs (64.5%), followed by Master’s degree programs (35%), and Doctoral degree programs (0.5%). A limitation of the sampling is self-election; therefore, we can only speculate that the data collected represent the whole Slovenian student population [16].

### 2.2. Materials and Instruments

**Depression**—The Patient Health Questionnaire (PHQ-9) [23]

To assess depression symptoms during the last 14 days, The Patient Health Questionnaire (PHQ-9) [23] was used. It is a widely used screening tool for depression in COVID-19 pandemic-related studies. It uses 9 items with DSM-V diagnostic criteria to assess depressive symptomatology on a 4-point Likert scale ranging from 0 (not at all) to 3 (nearly every day), with a total score ranging from 0 to 27. The higher scores are indicative of a higher presence of depressive symptoms and are clustered as follows: minimal (1–4), mild (5–9), moderate (10–14), moderately severe (15–19), and severe (20–27) [24]. An established cut-off point of 10 or above is used to classify participants as depressive symptomatic or not. Cronbach’s alpha of the PHQ-9 within the present study is 0.908. The PCA was conducted and, according to the results PHQ-9, is a unidimensional psychometric tool, which confirms the findings of its authors; in addition, the first component (eigenvalue = 5.199) explains 57.76% of variance.

The values of FIT indices for PHQ-9 are as follows: CMIN= 112.89; DF = 18; CMIN/DF = 6.27; CFI = 0.991; SRMR = 0.019; RMSEA = 0.047; and PCLOSE = 0.673.

**Anxiety—Generalized Anxiety Disorder questionnaire (GAD-7)** [25]

The Generalized Anxiety Disorder questionnaire (GAD-7) is a 7-item self-report measure to assess the severity of anxiety and its symptoms according to DSM-IV criteria. Participants rated how often they experienced anxiety symptoms in the past 2 weeks on a 4-point Likert scale from 0 (not at all) to 3 (almost every day). Total scores range from 0 to 21, with a cut-off score of 10 identifying instances of generalized anxiety disorder. The following cut-offs correlate with level of anxiety severity and scores ≥5, ≥10, and ≥15 are representative of mild, moderate, and severe anxiety symptom levels [26]. Cronbach’s alpha of the GAD-7 within the present study is 0.940. The PCA was conducted and, according to the results, GAD-7 is a unidimensional tool, which confirms the findings of its authors; in addition, the first component (eigenvalue = 5.173) explains 73.90% of variance

The values of FIT indices for GAD-7 are as follows: CMIN = 66.196; DF = 6; CMIN/DF = 6.62; CFI = 0.996; SRMR = 0.009; RMSEA = 0.049; and PCLOSE = 0.532.

**Stress—Perceived Stress Scale****-4 (PSS-4)** [27]

The PSS was designed to be an easy-to-understand and easy-to-report instrument for use with community samples with at least an intermediate level of schooling. The questions are quite general and, therefore, relatively free from population-specific content [27]. The items in the instrument ask respondents to report on their coping with various situations in the past month. The instrument has four items, two of which (items Q2 and Q3) have a reverse score. The response format is: 0 = never; 1 = almost never; 2 = sometimes; 3 = fairly often; 4 = very often. The scores are summed, and higher totals indicate higher levels of stress. The range of the scores is between 0 (the lowest score) and 16 (the highest score). Slovenian translation of the PSS-4 scale [28] was used in the study. The one-factor (latent variable) construct has a Cronbach’s alpha of 0.80, and is unidimensional, according to the test using PCA analysis (eigenvalue, 2.496; explained variance = 62.406%).

The values of FIT indices for PSS-4 are as follows: CMIN = 9.852; DF = 1; CMIN/DF = 9.852; CFI = 0.997; SRMR = 0.011; RMSEA = 0.062; and PCLOSE = 0.235.

**Resilience—10-Item Connor–Davidson Resilience Scale (CD-RISC-10)** [29]

To assess resilience among respondents, the 10-Item Connor–Davidson Resilience Scale (CD-RISC-10) was used. CD-RISC-10 is a self-report scale consisting of 10 items describing different aspects of resilience. The scale serves mainly as a measure of hardiness, with items corresponding to flexibility (1 and 5), sense of self-efficacy (2, 4 and 9), ability to regulate emotion (10), optimism (3, 6 and 8) and cognitive focus/maintaining attention under stress (7). Items are scored on a five-point scale ranging from 0 (statement is not at all true) to 4 (statement is true nearly all the time), with a total score ranging from 0 to 40. Higher scores suggest greater resilience and lower scores suggest less resilience, or more difficulty in bouncing back from adversity. Cronbach’s alpha of the CD-RISC 10 within this study is 0.885. The PCA was conducted and, according to the results, the CD-RISC-10 is a unidimensional tool, which confirms the findings of its authors; in addition, the first component (eigenvalue = 5.013) explains 50.132% of variance.

The values of FIT indices for CD RISC-10 are as follows: CMIN = 176.791; DF = 30; CMIN/DF = 5.893; CFI = 0.984; SRMR = 0.027; RMSEA = 0.046, and PCLOSE = 0.851.


**Satisfaction with Online Study Scale (SAT-5)-5**


The Satisfaction with Online Study Scale (SAT-5) is rooted in flow theory [30]. The measurement encompasses a 7-point Likert scale ranging from 1 = “strongly disagree” to 7 = “strongly agree,”, with a total score ranging from 5 to 35; higher scores indicate higher perceived satisfaction. Cronbach’s alpha of the SAT-5 within this study is 0.881. The PCA was conducted and, according to the results, the SAT-5 is a unidimensional tool and the first component (eigenvalue = 3.399) explains 67.971% of variance.

The values of FIT indices for SAT-5 are as follows: CMIN = 29.631; DF = 3; CMIN/DF = 9.877; CFI = 0.996; SRMR = 0.015; RMSEA = 0.062; and PCLOSE = 0.148.

### 2.3. Procedure

A cross-sectional study using a set of previously tested instruments and ad hoc questions created by the authors was chosen as a method to gain insight into various health and socio-demographic aspects of Slovenian post-secondary students affected by COVID-19-induced closures, and the suspension of educational activities, at tertiary educational institutions. From the gathered data, we then proceeded to extract the variables needed to address the present study’s objective: depression (PHQ-9), anxiety (GAD-7), stress (PSS-4), satisfaction with online studies (SAT-5), resilience (CD-RISC-10), and gender.

All constructs of interest were handled in a similar way. Because end sums of all items in the constructs were of interest after an initial data screening, all respondents with missing data were deleted from the poll. Each variable was screened for measures of central tendencies, skewness, and kurtosis. Cronbach’s alphas were calculated as a measure of reliability. Before proceeding to Exploratory Factorial Analysis (EFA) and Confirmatory Factorial Analysis (CFA), a sample was divided into two subgroups based on random selection, a procedure performed using a built-in algorithm of the IBM SPSS^®^ statistical package. EFA was conducted on a sample of 2332 (49.8%) and CFA on a sample of 2339 (50.2%) respondents, respectively. After an initial check of the data matrix using KMO and Bartlett’s tests, PCA with Direct Oblimin rotation and unidimensionality of constructs was explored. Because all constructs met this criterion, we proceeded with CFA analysis using SPSS AMOS^®^ software.

Data collection was conducted through a self-reported survey as a part of a large cross-sectional study to determine mental health status and factors which may influence postsecondary students in Slovenia. The prevalence of stress, depression, anxiety, resilience, and satisfaction with online study were selected as correlates in the study. The study took place between 9 February and 8 March 2021 on the whole territory of the Republic of Slovenia.

Participants were recruited online; the research was conducted through a web-based survey platform (https://www.1ka.si/) (accessed on 9 March 2021). Simple random sampling was used, and invitation letters to participate in the study were sent to all universities, private faculties, and student organizations with a request to forward the invitation to participate to all their students. To obtain as much feedback as possible, a reminder letter with the invitation to participate was sent to all addressees after one week, and then after another week, to those from whom we had not received any feedback. Participants were informed about various aspects of the study, including their rights to voluntarily participate or withdraw from the study. They were informed of the assured anonymity of their answers and that the gathered data would be used for scientific research purposes only. Ethical approval to conduct the study was obtained from the National Medical Ethics Committee of the Republic of Slovenia (NMEC), Ministry of Health (No. 0120-48/2021/3).

The survey was conducted over a one-month period. After ending data collection, we received responses from 5999 full-time students, who partially or fully responded to the questionnaires. After deleting the data from respondents who did not provide full sets of responses, we ended with 4661 respondents. The population of students enrolled in tertiary education in Slovenia a year before the study was about 59,000 full-time students, so the present data collection achieved about a 10% sample [31].

The decision to include only respondents who provided all the data was rooted in finding that data imputation with means or the most numerous values would bias the results; this is because the sums of all items in each of the scales depends on the number of items in the scales, which ranged from 4 (PSS-4) to 10 (CD-RISC-10).

Of the 4661 respondents 1245 (26.7%) were male, 3379 (71.5%) were female, and 36 (0.8%) stated “other”. The included constructs had the following missing data: anxiety—ANX (valid: 5403, missing: 596); stress—STR (valid: 4761, missing: 1238); depression—DEP (valid: 5386, missing: 613); resilience—RES (valid: 4744, missing: 1255); satisfaction—SAT (valid: 5386, missing: 613).

### 2.4. Statistical Analysis

We choose to perform analyses on the differences between respondents identifying as male and female. Persons who did not respond to this question (*n* = 1) or identified as other (*n* = 36) were excluded from analyses. The rationale to conduct this analysis was the reported differences between genders for all constructs, except SAT-5, in previous studies.

To assess the connection between satisfaction with online study and other constructs, we divided respondents in three to SAT scores. In the first group were students who were the least satisfied (cut-off SAT ≤ 13; *n* = 1678; 36%); in the second middle group were students in the range between cut-off SAT ≥ 14 and ≤19 (*n* = 1235; 26.9%); and in the third group were participants reporting the highest levels of satisfaction (cut-off SAT ≥20; *n* = 1730; 37.1%). The unequal size of the three groups based on the level of reported satisfaction is the result of the decision not to make a cut in the middle of the values of the variables. The comparative analysis between lower and upper SAT scores groups is provided.

Tests of normality were conducted in order to chose appropriate statistical tests for the data analysis. The data were not normally distributed; therefore, non-parametric tests were utilized due to being a more robust technique.

Spearman’s rho correlation coefficient was chosen to assess the measure of association. Power analysis revealed that the sample size allowed the identification of significant statistical differences in effect sizes below the 0.1 level.

The raw data and sums were analyzed; therefore, no intervention was made to correct the distribution or similar processes, or to correct for missing data. Power analysis was performed on the summed data. The data were processed using the statistics program IBM SPSS v. 27.0 and IBM AMOS v. 27.

## 3. Results

### 3.1. Corelations between Constructs

The aim of the correlation analysis was to find connections between constructs describing mental health (Depression (DEP), Anxiety (ANX), and Stress (STR)) and their association with resilience (RES) and satisfaction with online study (SAT). In Table 1, it was revealed that correlations between DEP, ANX and STR are high and can be recognized as almost inseparable. All three negatively correlate with RES and SAT, which, themselves, are weakly correlated (Table 1).

From the results, it can be interpreted that satisfaction with online study influences resilience. Both aforementioned constructs lower stress, even if resilience has a somewhat greater influence than satisfaction, and both are negative predictors of depression and anxiety. In absolute terms, it can mean that measures toward improving the mental health of students cannot be oriented only toward helping them, by means of psychiatry and psychology, with anxiety and depression, because these can be regarded as symptoms. It is plausible to make significant efforts to build resilience, which may be in the hands of psychology; lastly, but not of least importance, efforts can be made to improve satisfaction with study, which is the responsibility of postsecondary institutions.

### 3.2. Differences between Genders

In addition, we were interested in the differences between genders. The differences between genders (*n* = 4624) and the test statistics between genders are presented in Table 2 and Table 3.

The differences between genders are clearly seen for all constructs except SAT, and are statistically significant: ANX: female (M ± SD = 11.04 ± 6.48), male (M ± SD = 8.54 ± 6.29); DEP: female (M ± SD = 11.86 ± 7.26), male (M ± SD = 9.65 ± 6.86); STR: female (M ± SD = 8.26 ± 3.25), male (M ± SD = 7.12 ± 3.30); RES: female (M ± SD = 22.95 ± 7.23), male (M ± SD = 25.58 ± 7.35); and SAT: female (M ± SD = 17.91 ± 7.77), male (M ± SD = 17.91 ± 7.79).

From the analysis of the means, it is possible to reveal that there are significant differences between genders, where female gender was impaired to a larger extent for all constructs, except SAT.

### 3.3. Differences between Students Unsatisfied with Online Study and Those Who Were Satisfied with It

The differences in constructs between the lower (SAT < 13; *n* = 1678; 36%) and upper third (SAT > 20; *n* = 1730; 37.1%), according to the SAT scores, are statistically significant and large (sensu Cohen) for all constructs (Table 4 and Table 5).

## 4. Discussion

The COVID-19 pandemic has exposed students to numerous stressful events that can affect their mental health. Both foreign and domestic research suggests that mental health has deteriorated during the pandemic. Research shows, quite consistently, that it is young people who are among the most affected [13,15,17,18].

The results of the present study suggest that there is a strong association between depression, anxiety, and stress, and that they are difficult to distinguish from each other. The close associations between the aforementioned mental health constructs are not surprising, as stress has a significant impact on the development of anxiety and depression disorders [32,33], and the comorbidity of depression and anxiety often occurs. For example, a worldwide survey reported that 45.7% of people with major depression also experienced a lifetime of one or more anxiety disorders [34]. Nevertheless, it does raise concerns, as the comorbidity of these two disorders requires a more complex treatment than one alone [35,36].

To curb the spread of the virus, several measures have been taken that have severely interfered with the way of life as we knew it before. The pandemic and accompanying measures to prevent the spread of the coronavirus have also radically affected the field of education, throughout all the different layers of the educational system [5]. Most educational processes took place remotely [4,37], a situation applying mostly to tertiary education. Feedback from professors, student organizations, and finally, the students themselves suggests an increase in mental distress in the student population [37].

Furthermore, a negative correlation between depression, anxiety, and stress with psychological resilience and satisfaction with distance learning was found. Satisfaction with distance learning was also observed to have a significant impact on the levels of depression, anxiety, and stress, as well as on psychological resilience. These correlations are in line with previous research reporting that students’ poor mental health had a negative impact on their academic achievements, and vice versa [37,38,39]. During the COVID-19 pandemic, researchers have observed a positive correlation between delays in the completion of academic activities due to COVID-19 and levels of anxiety symptoms reported by students [40]. Moreover, the results of the present study also suggest that satisfaction with distance learning affects psychological resilience, and both, together, can reduce symptoms of depression and anxiety. The impact of psychological resilience on depression, anxiety, and stress during the COVID-19 pandemic has already been confirmed by research in China within the general population [41], among healthcare professionals [42], and among health science students [43]. Research also confirms the need for increased support for the student population, as identified in a study [44].

The present study uncovered large gender differences. Female students were found, in all respects—depression, anxiety and stress—to be significantly more burdened, while at the same time, reporting lower levels of psychological resilience compared to male students. This was observed despite the fact that there is no difference between the genders with regard to satisfaction with distance learning; this means that the effect of study satisfaction associated with depression, anxiety, stress, and psychological resilience is likely to be greater in men. Although the greater psychological vulnerability of female compared to male students was already echoed in previous research [12,13,15,17,45], to the best of our knowledge, no research has observed that male students appear to be more vulnerable to mental health difficulties when dissatisfied with distance learning during the COVID-19 pandemic.

One of the limitations of the present study was the absence of a pre-COVID-19 dataset; if at our disposal, this would provide a greater insight into the degree of impact that satisfaction with distance learning had on depression, anxiety, stress, and resilience. Moreover, the lack of a pre-COVID-19 dataset also resulted in a limited understanding of the gender differences that were observed in the present study.

Future research should focus on more in-depth exploration of the relationship between satisfaction with distance learning and psychological resilience with regard to stress, anxiety, and depression, by also including additional demographic variables. Moreover, it should also explore specific COVID-19 related variables (e.g., employment loss, nursing family members, and social isolation) and more specific distance-learning variables (e.g., inappropriate technical equipment, internet difficulties, and a lack of study space) in relation to postsecondary students’ mental health outcomes. This would enable researchers to identify which factors negatively impact students’ mental health can be quickly addressed, and which would need a more systematic, long-term approach.

## 5. Conclusions

The present results are highly significant, as they not only confirm the associations found by various other researchers, but also enable us to explore the associations of the variables with each other at once. More specifically, by including all the variables it was discovered that for students’ mental health, it is not only psychological resilience or satisfaction with distance learning in isolation that are important. Rather, it was suggested that the simultaneous presence of satisfaction with distance learning and psychological resilience can work as a protective factor for postsecondary students’ mental health during the COVID-19 pandemic. Moreover, while both play an important role, the results also highlighted that a tailored approach might be needed for male and female students, by placing more focus on the construct which appears to play a greater role in developing mental health difficulties. At this point, we note that in addition to the role of heath institutions, the competencies of post-secondary institutions also play an important role in their students’ mental health, as they can increase satisfaction with distance learning, strengthen psychological resilience and indirectly reduce stress, anxiety and depression. Students, especially those who are female, need greater psychological resilience to combat negative mental states and their unwanted psychological consequences.

## Figures and Tables

**Table 1 ijerph-19-07024-t001:** Means, standard deviations, and correlations between constructs. Spearman’s rho is reported (N = 4661; N_L_ = 1678; N_U_ = 1730).

	ANX	DEP	STR	RES	SAT	Mean	SD
ANX	--					10.39	6.53
DEP	0.816	--				11.29	7.23
STR	0.675	0.707	--			7.96	3.31
RES	−0.468	−0.483	−0.603	--		23.63	7.37
SAT	−0.443	−0.439	−0.357	0.254	--	17.91	7.77

Note: All correlations are significant at 0.001 level (2-tailed).

**Table 2 ijerph-19-07024-t002:** Values of means, distribution of constructs, and differences between genders.

		*n*	Mean	Std. Deviation	Std. Error	95% CI for Mean	
						Lower Bound	Upper Bound
ANX	Female	3379	11.04	6.48	0.112	10.82	11.25
	Male	1245	8.54	6.29	0.179	8.19	8.89
	Total	4624	10.36	6.52	0.096	10.18	10.55
DEP	Female	3379	11.86	7.26	0.125	11.61	12.10
	Male	1245	9.65	6.86	0.195	9.27	10.04
	Total	4624	11.26	7.22	0.106	11.05	11.47
STR	Female	3379	8.26	3.25	0.056	8.15	8.37
	Male	1245	7.12	3.30	0.094	6.94	7.31
	Total	4624	7.95	3.30	0.049	7.86	8.05
RES	Female	3379	22.95	7.23	0.124	22.71	23.20
	Male	1245	25.58	7.35	0.209	25.17	25.99
	Total	4624	23.66	7.35	0.108	23.45	23.87
SAT	Female	3379	17.91	7.77	0.134	17.65	18.18
	Male	1245	17.91	7.79	0.221	17.48	18.35
	Total	4624	17.91	7.77	0.114	17.69	18.14

Note: N *_n_*
_total_ = 4624; *_n_*
_female_ = 3379; *_n_*
_male_ = 1245.

**Table 3 ijerph-19-07024-t003:** Test statistics between genders.

	Mann–Whitney U	Asymp. Sig. (2-Tailed)	Eta-Squared	CI_L_	CI_U_
ANX	1,635,789.0	>0.001	0.029	0.020	0.039
DEP	1,732,506.5	>0.001	0.018	0.011	0.027
STR	1,693,845.5	>0.001	0.023	0.015	0.033
RES	1,660,424.5	>0.001	0.025	0.017	0.035
SAT	2,098,303.0	0.899	0.000	0.000	0.000

Margins for interpretation of effect sizes provided as Eta-squared are: 0.01 = small effect; 0.06 = medium effect; 0.14 = large effect.

**Table 4 ijerph-19-07024-t004:** Values of means, distribution of constructs, and differences between students satisfied and unsatisfied with online education.

	*n*	Mean	SD	Std. Error	95% CI for Mean	
					Lower Bound	Upper Bound
ANX-L1	1678	13.58	5.895	0.144	13.30	13.86
ANX-U	1730	7.53	6.194	0.149	7.24	7.82
DEP-L	1678	14.79	6.949	0.170	14.46	15.13
DEP-U	1730	8.21	6.648	0.160	7.90	8.53
STR-L	1678	9.28	3.201	0.078	9.13	9.43
STR-U	1730	6.76	3.213	0.077	6.61	6.91
RES-L	1678	21.56	7.794	0.190	21.19	21.94
RES-U	1730	25.61	7.019	0.169	25.27	25.94
SAT-L1	1678	9.62	3.029	0.074	9.47	9.76
SAT-U	1730	26.17	4.144	0.100	25.97	26.36

**Table 5 ijerph-19-07024-t005:** Test statistics of differences between lower (SAT < 13; *n* = 1678; 36%) and upper group of students (SAT > 20; *n* = 1730; 37.1%) according to satisfaction with online study (SAT).

	Mann–Whitney U	Z	Asymp. Sig. (2-Tailed)	Eta-Squared	CI	CI	Interpretation Sensu Cohen
ANX	701,278.5	−26.102	0.000	0.200	0.177	0.222	Large
DEP	713,841.5	−25.650	0.000	0.189	0.167	0.212	Large
STR	834,695.5	−21.495	0.000	0.133	0.113	0.154	Large
RES	1,015,406.5	−15.127	0.000	0.069	0.053	0.086	Large

Margins for interpretation of effect sizes provided as Eta-squared are: 0.01 = small effect; 0.06 = medium effect; 0.14 = large effect.

## Data Availability

Supporting results can be found at: https://www.nijz.si/sl/ukrepi-na-podrocju-obvladovanja-siritve-covid-19-s-poudarkom-na-ranljivih-skupinah-prebivalstva (accessed on 11 October 2021).

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
