# Peer review of "Perceived Satisfaction with Online Study during COVID-19 Lockdown Correlates Positively with Resilience and Negatively with Anxiety, Depression, and Stress among Slovenian Postsecondary Students"

_ijerph, 2022, doi:10.3390/ijerph19127024_

Round 1

Reviewer 1 Report

The manuscript presents a study that aim “to assess the prevalence of stress, anxiety, and depression among university students during the COVID-19 pandemic; to examine the association of resilience with symptoms of stress, depression, and anxiety during the pandemic and to examine the role of satisfaction with online distance learning in relation to psychological resilience, stress, anxiety, and depression. Although this is a very important and relevant issue to investigate, the paper has several problems of structure and information presentation. So, the paper need profound reformulation before being published.

For instance:

  • The introduction is the part of the manuscript that is better written but more consistent literature review is needed.
  • The Study objectives and hypotheses are very confused. This part needs to be more objective and in accordance with the analysis performed.
  • The Method part has several mistakes, for example the authors name this part as “Materials” that should be Method, with sections: participants/sample; materials and instruments (with construts); procedure and statistical analysis. The authors present a point “2. Constructs”, inside the materials and methods??? This refers to instruments. No details about ethical procedure are presented. Also, some information’s on data analysis are presented in the procedures, that should be on the statistical analysis.
  • Results are not in accordance with the objectives, for instance differences between genders are presented, and this is not formulated in the objectives. The authors presented in a very global and not detailed the results.
  • The discussion is very poor and not in accordance with the goals of the study and the results presented.

Author Response

Dear Sir or Madam,

thank you for your feedback and advice on how to improve our article. 

As instructed, we have made great revisions in terms of introduction by adding additional references and making it more consistent; the objective of the study was made more clear; corrected the method section (revised the order of subheadings, renamed them) and expended the discussion. As the objectives of the study were made more clear the results and discussion are now aligned with it as well.

With kind regards

Reviewer 2 Report

A very current, complex, well-conducted study, with a solid statistical analysis. 

For the first paragraph of the discussion we suggest highlighting some bibliographic sources to support the statement.

From the bibliography, it is NOT very clear to me what position 11 represents. Maybe more clarifications are needed.

1. In short, the article aims to complete the specialized studies that highlight the connection between the influence of satisfaction with distance learning on the correlates of mental health in Covid-19.

2. The manuscript’s strengths are: - the topicality of the research problem and the correlation targeted by the research; - quality of Presentation; - Scientific Soundness; - the description of the research and the presentation of the results, in a clear, correct and complete manner;

 I did not suggest major revisions, only minor ones.

These are:

1. For the first paragraph of the discussion we suggest highlighting some bibliographic sources to support the statement.

2. From the bibliography, it is NOT very clear to me what position 11 represents. Maybe more clarifications are needed.

Author Response

Dear Sir or Madam,

thank you for your feedback and suggestions on how to improve our paper.

We put additional references in the discussion and also corrected the error in the citation of the 11th source (now 18th). 

With kind regards

Reviewer 3 Report

Is a reports quasi-experimental study.

The objectives of the manuscript are clearly enumerated.

The literature review is comprehensive.

The references included are relevant for the subject under study, and show  references from the last 5 years.

There is concordance between the objective and the methods used.

The description of the methodology it  clear and adequate. Presents data from statistical analysis in the results. The ethical issues is adequately considered.  The discussion correlates with the presented data and takes published literature into account. The manuscript presents the practical implications of the study. The limitations of the study  are presented. 

Congratulations !

Author Response

Dear Sir or Madam,

we would like to thank you for your feedback on out article, it was most kind of you giving us additional motivation! :)

With kind regards

Reviewer 4 Report

See attached review document for suggestions and feedback.

I go the sense that you are publishing other articles from this study, but this was not clearly stated.

Author Response

Dear Sir or Madam,

thank you for your feedback and suggestions on how to improve our paper.

We have phrased it more clearly that the data used in the present study is part of a larger study from which additional papers have been written. Of course we also included all your gramatical suggestions as they are always more than wellcome to a non-native speaker.

With kind regards

Reviewer 5 Report

Perceived satisfaction with online study during COVID-19 lockdown correlates positively with resilience and negatively with anxiety, depression, and stress among Slovenian postsecondary students 

This study examines the influence of satisfaction with distance learning on the correlates of mental health in Covid-19 pandemic, via an online cross-sectional study and with a large sample of 4661 postsecondary students. Five validated instruments - PHQ-9 (depression), GAD -7 (anxiety), PSS-4 (stress), CD -RISC-10 (resilience) and SAT -5 (satisfaction with online study) were used.

The findings suggest high correlations between anxiety, depression, and stress, and satisfaction with online learning and psychological resilience were negatively correlated with anxiety, depression, and stress. Satisfaction with online learning was also negatively correlated with psychological resilience. In addition, females showed higher levels of vulnerability to anxiety, depression, and stress and exhibited lower levels of psychological resilience than males. 

A general observation is that the survey research was properly contacted, and the report is well presented in all parts. The measurement part is written, and all necessary information is provided.

Concerning the statistical analysis, merely descriptive, correlations and group differences (genders) is presented. This simple statistics is the weak point of the paper, which however could be improved.

-The sample is very large and parametric t-test could be used instead or along with the Mann Whitney test. Please provide (or discuss) and compare results from the parametric test.

For correlation analysis parametric coefficient was used, and restricted to this only, very little is learned about the relations among the variables under investigation.

I would suggest, since data are available, to show some results from multiple regression analysis in order to investigate concomitant effects of the predictor variables. This will improve the paper, since bivariate correlational analysis is methodologically poor.

Author Response

Dear Sir or Madam,

thank you for your feedback and suggestions on how to improve out paper. 

In order to respond in a clear fashion we put your feedback in quotation marks, which is then followed by our response. 

"The sample is very large and parametric t-test could be used instead or along with the Mann Whitney test. Please provide (or discuss) and compare results from the parametric test."

We agree with the reviewer that we have large sample, however, a number of tests used before the final choice to use non-parametric statistics was made (Shapiro Wilks, Kolmogorov Smirnov, Q-Q plots, etc.) and later on homogeneity of variance show, that our variables are not normally distributed therefore more robust techniques were applied. All tests were performed with assumption of normality as well, and the differences were found to be statistically significant in all cases as was already shown by non-parametric tests. According to trends in reporting statistics we were more concerned in assessment of effect sizes. We also added the reasoning for our use of non-parametric tests in to the Methods section under the Statistical analysis. 

"For correlation analysis parametric coefficient was used, and restricted to this only, very little is learned about the relations among the variables under investigation."

All the instruments used in our analysis were intented to be used as end sums to predict the anxiety, depression, etc., therefore we were not so much interested in contribution of individual items (Kline, 2010) but with latent variables (constructs) expressed as sums of responses.

"I would suggest, since data are available, to show some results from multiple regression analysis in order to investigate concomitant effects of the predictor variables. This will improve the paper, since bivariate correlational analysis is methodologically poor."

We appreciate the suggestion; however, we did not perform a regression analysis because it is hard to define outcome variable to be regressed. It can be realized that there are three latent variables worsening mood and two (RES, SAT) which works in positive way. It means that they work as a complex influencing each other, that was the reason for correlational analysis.

With kind regards

Reviewer 6 Report

The submitted manuscript covers a topic of special relevance to the scientific community. It is well designed and structured. However, the authors have to take into account the following observations to improve the quality of the study:

-The theoretical framework should be increased. The state of the question must be delved into. Reading the following manuscript on Covid-19 is recommended:

https://doi.org/10.3390/su13105452

-Finally, the discussion section should be worked on in greater depth. The authors have to discuss the results achieved with the previous literature. It is not enough to do it superficially. Likewise, the study lacks implications or prospective derived from such findings. It would be pertinent to add this information to the manuscript.

Author Response

Dear Sir or Madam,

thank you for your feedback, suggestions on how to improve our paper and the paper you suggested for us to read (it was a really good one - thank you).

We have increased the theoretical framework of the paper as well as our discussion. We included the suggestions for the future research and implications of the results. 

With kind regards